# Feasibility of Social-Network-Based eHealth Intervention on the Improvement of Healthy Habits among Children

**DOI:** 10.3390/s20051404

**Published:** 2020-03-04

**Authors:** José Alberto Benítez-Andrades, Natalia Arias, María Teresa García-Ordás, Marta Martínez-Martínez, Isaías García-Rodríguez

**Affiliations:** 1SALBIS Research Group, Department of Electric, Systems and Automatics Engineering, University of León, Campus of Vegazana s/n, León, 24071 León, Spain; 2SALBIS Research Group, Department of Nursing and Physiotherapy Health Science School, University of León, Avenida Astorga s/n, Ponferrada, 24401 León, Spain; narir@unileon.es; 3SECOMUCI Research Groups, Escuela de Ingenierías Industrial e Informática, Universidad de León, Campus de Vegazana s/n, C.P. 24071 León, Spain; mgaro@unileon.es (M.T.G.-O.); isaias.garcia@unileon.es (I.G.-R.); 4University of León, Campus of Vegazana s/n, León, 24071 León, Spain; mmartm16@estudiantes.unileon.es

**Keywords:** eHealth, mHealth, healthy habits, obesity, physical activity

## Abstract

This study shows the feasibility of an eHealth solution for tackling eating habits and physical activity in the adolescent population. The participants were children from 11 to 15 years old. An intervention was carried out on 139 students in the intervention group and 91 students in the control group, in two schools during 14 weeks. The intervention group had access to the web through a user account and a password. They were able to create friendship relationships, post comments, give likes and interact with other users, as well as receive notifications and information about nutrition and physical activity on a daily basis and get (virtual) rewards for improving their habits. The control group did not have access to any of these features. The homogeneity of the samples in terms of gender, age, body mass index and initial health-related habits was demonstrated. Pre- and post-measurements were collected through self-reports on the application website. After applying multivariate analysis of variance, a significant alteration in the age-adjusted body mass index percentile was observed in the intervention group versus the control group, as well as in the PAQ-A score and the KIDMED score. It can be concluded that eHealth interventions can help to obtain healthy habits. More research is needed to examine the effectiveness in achieving adherence to these new habits.

## 1. Introduction

According to the World Health Organization, one of the main health problems among children and young people in the 21st century worldwide is obesity [1]. The global prevalence of obesity and overweight in the 5–19 year old population has increased dramatically between 1975 and 2016 [2].

In Spain, according to data from the latest National Health Survey (ENSE) [3], the percentage of young people between 2 and 17 years of age who are overweight is 18.26%, while the percentage of young people with obesity in this age range is 10.3%). The presence of overweight or obesity in the child and adolescent population is associated with various health problems, such as cardiovascular and metabolic disorders, pulmonary complications, gastrointestinal or musculoskeletal disorders [4]. There has also been evidence of a worse quality of life [5] and lower self-esteem among young people [6], as well as problems of marginalization and social isolation [7]. The development of overweight or obesity is determined by a combination of different factors such as genetics, metabolism, environmental factors or individual behaviour, as well as socioeconomic and socio-cultural aspects, although it has been found that inappropriate eating habits and/or lack of physical activity can be considered the main modulating factors and, therefore, are the elements that are usually affected by interventions that address this problem [8].

Traditional interventions to address the problem of excess weight have focused on the individual, seeking to encourage healthy and responsible behaviour in the individual. In recent years, the study of factors related to this type of problem has also focused on other types of determining factors, such as the economic, political, environmental and social environment in which the individual is subject to [9]. A number of studies have shown the benefits of interventions for the prevention and treatment of obesity in child population. Some of these studies focus on the promotion of physical exercise, good eating habits, or both, but the certainty on the reported alteration of the studied variables is usually low [10]. There is, therefore, a need to improve our knowledge into this type of research.

With the development of information and communications technologies, as well as the proliferation of mobile devices, the possibilities for intervention in the field of health, and in particular in the promotion of physical activity and healthy eating habits, have multiplied. The use of mobile devices makes it possible to implement applications that connect users in virtual communities in order to promote changes in health-related behaviours, in an environment in which the relationships between users can be very beneficial in terms of the efficiency of the intervention [11]. There are several studies suggesting that eHealth interventions can change diet and physical activity in adolescents [12].

The effectiveness of eHealth systems has been studied in the systematic review by Lau et al. [13], who studied tailored eHealth interventions in overweight and obese adults. They concluded that personalised eHealth interventions is a suitable approach for overweight and obese adults to reduce weight, BMI, waist circumference and blood pressure. Other authors such as Thompson et al. studied the efficiency of an eHealth application in the prevention of obesity in a group of 80, 8 to 10-year-old African American girls at risk of obesity. In this study, it was concluded that Internet-based obesity prevention programs may be an effective channel for promoting a healthy diet and physical activity in youths at-risk of obesity [14]. Therefore, this research aims to test the effectiveness of an eHealth application in improving the BMI age-adjusted percentile, physical activity and eating behaviours of adolescents.

Based on all of the above evidence, it was decided to create an eHealth application called “SanoYFeliz” (“Healthy&Happy”) aimed at improving the acquisition of healthy habits by the adolescent population. SanoYFeliz is a tool to assist in the prevention of juvenile obesity. This tool consists of a web application made with a responsive design that allows the visualization on computers, mobile devices and tablets with the same quality. Through this application, the individual is able to create a profile with his or her data, on the basis of which he/she obtains a series of individualized recommendations and advice related to the alteration in nutritional and sports habits, thanks to the use of artificial intelligence algorithms and semantic technologies. The user is able to contact people who have a similar profile, creating a relationship of mutual support between both individuals. It also has a section for direct contact with health professionals and researchers if necessary, as well as forums and open communities that encourage self-monitoring and self-motivation among the users of the application.

This manuscript presents the feasibility study of the “SanoyFeliz” project. The changes made in the habits of the adolescents before and after using the SanoyFeliz application are presented, comparing this information with the one obtained from a control group. This feasibility study provides insight into factors that contribute to the development and use of eHealth applications among children to improve their habits, from unhealthy to healthy ones.

## 2. Materials and Methods

A pre-post-test design was used, gathering data by conducting a survey among the children before using the “SanoyFeliz” eHealth application and 14 weeks after. This study was approved by the Ethics Committee of Universidad de León (ETICA-ULE-028-2018) and accepted by the Ministry of Education of Castilla y León.

### 2.1. Participants

The sample consisted of 139 adolescents in the first and second year of Compulsory Secondary Education (CSE) at the the “Colegio Leonés” school and the control group consisted of 91 adolescents in the first and second year of Compulsory Secondary Education (CSE) at the “Colegio Maristas San José” school. Both schools are located in the city of León, in the province of León (Spain). In order to participate in the study, parents signed an informed consent form. The experience consisted of a pre-post quasi-experimental study using the technique of intentional or convenience sampling.

### 2.2. Study Design

The data used as a starting point for this work were obtained during the implementation of a project, in order to promote physical activity and healthy eating habits in adolescents. The levels of healthy eating habits and physical activity were studied and related to the use of the application. The main objective of this research was to demonstrate the feasibility of the described eHealth web application in an intervention to improve the physical activity and eating habits of the adolescent population and thus prevent overweight and obesity. The variables object of study consisted of the students’ body measurements, thanks to which measures such as the Body Mass Index and the age-adjusted percentile BMI were obtained. These measures are fundamental to know the effectiveness of the application in the alteration of the students’ body composition. Profiles of eating habits and physical activity were also obtained, the latter thanks to the KIDMED and PAQ-A questionnaires respectively. These are validated questionnaires have been used in several research studies [15,16,17,18].

The intervention aims to promote in adolescents the self-management of their own health, encouraging changes in behavior and the development of skills and psychosocial capabilities of these young people in the adoption of healthy habits and behaviors. It was carried out during the months of October and December 2019, with the aim of testing the functionalities of the mobile application developed, as well as gathering data on the participation and use given to the system by adolescents. Among the functionalities of the mobile application, the creation of an online social network of contacts in which only the young people who are part of the program participate, under the supervision of the researchers, stands out. Students of the 1st and 2nd year of Compulsory Secondary Education (CSE), between the ages of 12 and 15 participated in the experience. The student was free to participate in the application without further interference, in terms of encouraging participation by the research team, other than the weekly reminders in the classroom carried out by the physical education teacher.

The mobile application allowed each participant to have their own profile by means of a user name and a password. Its operation is similar to that of the Facebook social network but is only made up of the participants in the intervention and the profiles of the system’s administrators and moderators (a technician and a researcher from the socio-health field). Each user was able to connect with any of the others, but for this connection to be effective, the recipient of the request had to accept the establishment of the connection. The system also offered suggestions for connection on a random basis, and the user could decide whether to accept the suggested request or not.

Users could add “entries” in the application, consisting of an optional text, by photos and links, also indicating whether the publication was intended to be seen by any other user or restricted so that only users connected to the one who created entry could see it. All entries were processed and filtered by the moderators to avoid inappropriate content. As with the Facebook social network, other users could respond to posts by posting messages or reacting with a “like” to the post.

Another feature of the application was to create events (by entering the title, description and date of the event), which had to be related to sports activities or healthy eating topics. Other users could, in turn, indicate their intention to attend these events. The actual attendance check was not implemented for the pilot test.

Each time the user connected to the application, he or she was offered advice on healthy eating or related to physical activity, chosen from those stored in the database. Every activity carried out in the application (creating entries, activities, responding to other users, giving “likes”, etc.) was acknowledged with a system of virtual rewards called “bieneSTARS” in order to increase commitment to the intervention. There were no other rewards associated with participation. Further details of the application can be found in the conference paper presented at the HEALTHINF 2020 conference [19].

The control group only had access to the public part of the website, they did not have a username and password to access the platform. On the other hand, the intervention group had full access to the application. The users of the application who have access to the restricted area (intervention group), can make use of all the functionalities of the application:Access to the social network: add friends, comment on different walls, give like to publications, create events and get points in the reward system (bieneSTARS).Personalized notifications: the application sends personalized notifications and advice about nutrition and physical activity. This is accomplished by using push notifications available on phones, tables and web browsers, as well as sending emails.

Meanwhile, visitors to the website (control group) can only view the front page, project information and access the blog that contains articles on nutrition and physical activity of approximately 1000 words.

In this way, it should be noted that the students in the intervention group had access to the web application on a daily basis, including all the functionalities, while the control group could only access the public part mentioned above (no access to the social networks, personalised notifications, reward systems, etc.).

### 2.3. Measurements

#### 2.3.1. Sociodemographic and Anthropometric Variables

The sociodemographic variable used was gender, which is of a dichotomous type, being able to take the “female” (F) or “male” (M) values.

The anthropometric variables gathered were weight and height. These variables were gathered at the beginning and at the end of the intervention. In accordance with the objectives of the research project in which this intervention was framed, the aim was to make the adolescents responsible for managing their own health, so that they took these measurements themselves under the supervision of their parents. Previously they were told how to obtain these measurements and a video tutorial was sent to them so that they could consult it whenever they wanted. Adolescents could take and record their measurements at any time using the mobile application, although the values used to calculate the body mass index (BMI) that was later used in the data analysis were the measurements recorded at the beginning and at the end of the experience. Once the value of the body mass index of each adolescent was calculated, its position in terms of percentile was found, taking into account gender and age, as calculated by the World Health Organization (WHO) [20], and subsequently classified into one of the following categories: “obesity”, “overweight”, “normal weight” and “underweight”. To make these calculations, the Anthro Plus tool [21] provided by the WHO itself through its website was used.

Using this data, a dichotomous variable known as “percentile alteration” has been generated to analyse whether there has been an alteration in the percentile at the end of the intervention. For this purpose, it is considered that all those who initially showed a percentile of less than 50, would improve their percentile the closer they got to 50, that is, by increasing between a first measurement and a second measurement. While, those who showed a percentile of more than 50 in the first measurement, would improve if it were reduced.

#### 2.3.2. Healthy Nutrition

In order to find out the eating habits of adolescents, the KIDMED [22] Mediterranean diet quality questionnaire was used, which consists of 16 questions with two possible answers (“yes” or “no”). Affirmative answers to questions related to healthy eating habits add one point, while affirmative answers to questions related to unhealthy habits subtract one point. The total score is obtained from the sum of all the results obtained in the different items, and can vary between 0 and 12 points.

The ranges proposed by the authors to interpret the result are as follows: a score equal to or less than 3 points is equivalent to “very low quality diet”, between 4 and 7 points is equivalent to “need to improve the eating pattern to adapt it to the Mediterranean model”, and 8 points or more would be “optimal Mediterranean diet”.

In order to facilitate the handling of the data in the analysis phase, the dichotomous variable “feeding level” has been created, which considers two levels. On the one hand, the “adequate” feeding level corresponds to the “optimal Mediterranean diet” (8 points or more) and on the other hand, the “inadequate” feeding level that includes the deficit categories (“needs to improve the feeding pattern to adapt it to the Mediterranean model” and “very low quality diet”). In this way, a categorization is achieved that identifies the individuals that should be affected by the intervention.

Through this score, a dichotomous variable called “KIDMED Improvement” has been generated with a value of 1 if there is alteration and 0 if there is no alteration from the first recorded response and the last one.

#### 2.3.3. Physical Activity

The level of physical activity of the adolescents was gathered using the PAQ-A questionnaire (Physical Activity Questionnaire for Adolescents) [23]. The questionnaire contains eight questions related to the type and amount of physical exercise carried out during the last seven days. Each of the questions has a score of between 1 and 5, with the final score of the questionnaire being the average of all of them. There is a ninth question used to find out if the adolescent has been ill during the last week, in which case the value obtained in the variable is not taken into account and that person must be excluded from the corresponding statistical analysis.

As in the case of eating habits, the dichotomous variable “level of physical activity” has been created from the scores and intervals used by Chen, Lee, Chiu, & Jeng (2008) [24], which considers “low activity” for scores of less than or equal to 2, “moderate activity” for those greater than 2 and less than 3, and “high activity” for scores equal to or greater than 3. The dichotomous variable created considered “adequate” physical activity for the categories of “moderate activity” and “high activity”, and “inadequate” physical activity for the category “low activity”. In this way, as in the case of eating habits, the aim is to characterise the individuals who should be affected by the intervention.

Using the PAQ-A score, a dichotomous variable called the “PAQ-A Improvement” has been generated with a value of 1 if there is alteration in physical activity and 0 if there is no alteration from the first recorded response to the last one.

### 2.4. Data Collection

On the first day of the experience, the researchers visited the school and met with the students in their classrooms to explain the procedure for gathering the anthropometric parameters: weight and height. The effective collection of these data was carried out by the students themselves at home under the supervision of their parents. Prior to the last day, a reminder was sent to the students to take their measurements again and write them down in the application.

The sociodemographic variables, date of birth, data on physical activity, eating habits and the friendship network of the adolescents were obtained by means of web questionnaires completed on the first and last day of the intervention during a classroom session with the presence of the researchers.

The information was processed on anonymized data using the tool described in Benítez et al. (2017) [25], respecting confidentiality in accordance with Constitutional Law 3/2018 of December 5th on the Protection of Personal Data and the guarantee of digital rights, and following the recommendations of the Spanish Data Protection Agency.

## 3. Results

### 3.1. Descriptive Study

#### Homogeneity of the Control and Intervention Groups

The first analysis, which should confirm that the samples to be tested (intervention and control groups) are similar, is that of sample homogeneity. This will indicate, with sufficient confidence, that the final change in outcome variables is due to exposure to the implemented intervention and not to other possible causes, since chance can explain the possible differences between the two initial samples. In this case of study, it is necessary to check the homogeneity for the following outcome variables: gender, age, body mass index, BMI age-adjusted percentile, fitness, KIDMED score and PAQ-A score.

The multivariate analysis of variance test was used to study the homogeneity between the control and intervention groups. For this purpose, the belonging to the intervention or control group has been used as an independent variable, and gender, age, body mass index (BMI), BMI age-adjusted percentile, fitness, KIDMED score and PAQ-A score were treated as dependent variables. No significant differences were found (p = 0.109, partial eta squared = 0.051). Therefore, it can be concluded that there is no contamination in the sample and that the intervention and control groups are homogeneous.

Table 1 shows the different measures compared below, in the intervention group (IG) and the control group (CG).

A number of parameters were obtained from the user interactions with the eHealth application in the intervention group: number of sessions initiated (interactions), friend requests (sent, accepted and rejected), number of posts, number of likes received, number of events and reward points acquired. Table 2 shows the total number, the average per user and the average per day of each of them.
Interactions: number of sessions initiated in the application throughout the intervention.Friends requests: number of friend requests sent, accepted and rejected.Posts: number of posts published by users.Likes: number of likes received in the different publications.Events: number of events created by users.Acquired Reward Points: number of reward points acquired through the “bieneSTARS” reward system.

### 3.2. Statistical Analysis

Suitable statistical tests were used to analyse the aforementioned variables according to the type of variable and its fit to the normal distribution. SPSS v.24.0 software was used for the statistical processing of the data obtained. The Multivariate analysis of variance was used to check that differences between outcome variables were significant, taking into account different covariates that could be confusing. For the use of this test, it is not necessary to check the normality of the distribution of the variables. The results obtained in relation to the variables of physical activity, eating habits, BMI and age-adjusted percentile after the intervention are presented below.

#### 3.2.1. Individual Study of Intervention and Control Group Results

Table 3 shows the statistical correlation analysis taken separately the intervention and control groups. In the case of the BMI percentile, a partition has been made in the sample. This partition is due to the fact that individuals with an initial percentile greater than 50% will improve their BMI if they lower their percentile value, while individuals with an initial percentile less than 50% are said to improve their BMI if they increase its value.

In the case of the intervention group, an alteration of BMI percentile is observed for both individuals with an initial percentile greater than 50% and lesser than 50%. This alteration is statistically significant in both cases. In the case of physical activity a slight worsening is observed, but with no statistical significance. Finally, the eating habits are significantly increased. In the control group, there is a significant worsening in the BMI percentile for children who have an initial percentile lesser than 50% while there is a slight, non-significant, alteration for children with an initial percentile greater than 50%. The physical activity significantly worsens and the eating habits present almost no change. These results indicate that the intervention may be beneficial for improving some of the studied variables, and for moderating the worsening of other ones. In the next section, a comparative study of the results obtained by the control and intervention groups is presented, including a multivariate analysis test for adjusting for covariates.

#### 3.2.2. Difference between the Intervention and Control Groups

In order to compare the results of the control and intervention group results, a measure of the alteration (or worsening) in the variables have been calculated for the BMI age-adjusted percentile, the PAQ-A score and the KIDMED score. In the case of the percentile, the individuals having an initial value greater than 50% are said to improve their percentile when they lower this value approaching this 50% value, while individuals having an initial percentile value under 50% are said to improve their percentile when they increase this value approaching the 50%. This way, a measure of the alteration in the BMI percentile is calculated for each individual. The PAQ-A and KIDMED scores alteration (or worsening) are obtained by direct subtraction between the last value and the first one.

The results obtained by this test showed that the differences in the alterations in BMI age-adjusted percentile, KIDMED score and PAQ-A score between the intervention and control groups were highly significant. For the alteration of BMI age-adjusted percentile and KIDMED score a *p*-value < 0.001 was obtained in both cases, for the alteration of PAQ-A score a *p*-value = 0.002 was obtained. In order to include possible covariates in the study, a multivariate analysis of variance test was performed by adding gender, age and initial BMI as covariates. After studying and adjusting for these covariates, the significances showed similar values (*p* < 0.01 for alteration of BMI age-adjusted percentile, *p* = 0.034 for alteration of KIDMED score and *p* = 0.016 for alteration of PAQ-A score). Figure 1 shows the estimated marginal means for these values.

## 4. Discussion and Conclusions

This study shows the feasibility analysis for an eHealth intervention on healthy physical activity and eating habits among adolescents. The objective was to test whether a web-based eHealth application including a social network could help to alter these habits and result in an alteration of measures like the BMI age-adjusted percentile of this population. The intervention was designed following the behavior change theory and some of the well-known behavior change techniques described in [26]. These techniques are based on providing adequate and personalized information to the individual, creating a network of peers for reinforcing healthy behaviors, encouraging users to perform healthy actions, offering rewards for good practices, etc. The analysis of the results of the study shows a significant increase in the age-adjusted percentile, and the interpretation of this result is confirmed when comparing this result with that obtained in the control group. The difference between adolescents who increased their percentile in the intervention group (72.8%) and those who increased in the control group (29.30%) is significant. This shows that the fact of making use of the application, and the intervention itself, seems to have brought about an alteration in the life habits followed by the students in the intervention group. An alteration in the eating habits has also been found in the intervention group, but not in the control group. The physical activity has slightly worsened in the intervention group, but with no statistical significance. On the contrary, this measure has significantly worsened in the control group. All these facts show that children in the intervention group significantly increased the BMI age-adjusted percentile and the KIDMED score (eating habits) when compared to the individuals in the control group. The physical worsened in terms of the PAQ-A score, but interestingly this worsening was only found to be statistically significant in the control group. This study does not attempt to demonstrate that eHealth interventions are better than more traditional interventions for this problem. However, our results coincide with those presented in the study by Champion et al. (2016), in which body measurements and physical and nutritional activity habits improve during the intervention and, a behavior change is observed [27]. There are a number of limitations in this study and, therefore, it cannot be confirmed that eHealth interventions are positive in improving post-intervention habits. Longitudinal studies would be needed to corroborate that the alteration of habits is maintained over the years. But the results obtained shows the feasibility of these kind of approaches. Overweight and obesity can lead to heart diseases, such as cerebrovascular disease, stroke, Alzheimer’s Disease and other neurodegenerative diseases, as well as diabetes and metabolic syndrome, among many others. Interventions based on the use of applications like the one in this study may help to reduce the prevalence of overweight and obesity in the adolescent population. This finding implies a reduction in health expenditure by the population, since if this prevalence is reduced, the health expenditure of diseases associated with overweight and obesity will also be reduced. This application is in an experimental phase, but, with a proper funding, it could be made public and accessible to everyone.

## Figures and Tables

**Figure 1 sensors-20-01404-f001:**
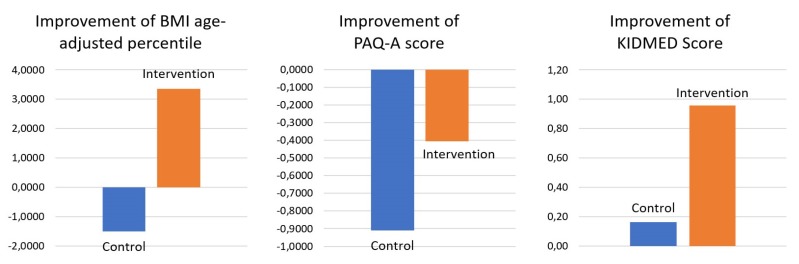
Estimated marginal means for the alteration of BMI age-adjusted percentile, PAQ-A score and KIDMED score in intervention and control groups for the model including covariates.

**Table 1 sensors-20-01404-t001:** Gender, age, BMI age-adjusted percentile, KIDMED score and PAQ-A score in intervention group (IG) and control group (CG).

	Mean CG	Mean IG	SD CG	SD IG	p
Gender	0.53	0.58	0.51	0.50	0.411
Age	12.77	12.63	0.62	0.59	0.081
BMI age-adjusted percentile	57.12	49.68	29.15	30.4	0.067
KIDMED score	7.31	7.06	3.24	2.50	0.522
PAQ-A score	2.98	2.75	0.90	0.97	0.078

**Table 2 sensors-20-01404-t002:** Measures obtained in the intervention group to evaluate the use of the application.

	Total	Average per User	Average per Day
Interactions	7696	58.75	80.17
Friend requests	1127	8.60	11.74
Accepted friend requests	181	1.38	1.89
Rejected friend requests	222	1.70	2.31
Posts	3722	28.41	38.77
Likes	4727	36.08	49.24
Events	107	0.82	1.12
Acquired Reward Points (BieneSTARS)	11215	85.61	116.82

**Table 3 sensors-20-01404-t003:** Changes in BMI age-adjusted percentile, KIDMED score and PAQ-A score in intervention group (IG) and control group (CG).

		Mean Pre	Mean Post	Median Pre	Median Post	Z	*p*
	BMI age-adjusted percentile ( >50 initial)	77.59	72.85	77.65	71.40	−5.394	0.000
IG	BMI age-adjusted percentile ( <50 initial)	22.94	25.57	24.40	25.60	−2.653	0.008
	PAQ-A Score	2.75	2.35	2.90	2.84	−0.666	0.505
	KIDMED Score	7.06	8.02	8.00	9.00	−5.960	0.000
	BMI age-adjusted percentile ( >50 initial)	78.09	77.49	82.10	81.65	−0.241	0.809
CG	BMI age-adjusted percentile ( <50 initial)	26.53	24.33	27.30	21.60	−2.421	0.015
	PAQ-A Score	2.98	2.06	3.05	2.60	−5.099	0.000
	KIDMED Score	7.31	7.47	8.00	8.00	−0.482	0.630

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
