# Peer review of "Feasibility of Social-Network-Based eHealth Intervention on the Improvement of Healthy Habits among Children"

_sensors, 2020, doi:10.3390/s20051404_

Round 1
Reviewer 1 Report
I was pleased to read your manuscript „ Feasibility of Social-Network Based e-Health Intervention on The Improvement of Healthy Habits Among Children”. This an interesting theme; the improvement of healthy habits among children issue is an actual theme and a concern for preventing human diseases and improving the personal wellbeing of the future generation. The idea of creating a social network application to develop a community, which collaborate and make exchange of information about a healthy life, thus improving eating and physical activity habits and self-efficacy in the child population is an interesting idea. The authors tested a self-created multi-platform web-based on a group of children with an age between 11 and 15 years old to acquire relevant data about their eating and physical activity habits. This paper has the potential to provide useful insights regarding how this kind of applications may be useful for the younger generation aware of the importance of practicing sport and a healthy diet. At this point, however, the paper is not yet able to realize that potential.
Therefore, I would suggest some changes in order to improve the manuscript. Below you will find my specific comment:
The term “e-Health” is used only in the title and on rows 68 and 74, in the rest of the paper you use the “eHealth”. Please be consequent.
Is not clearly specified the role of the control group. Abstract should be supplemented with information about the differences between the two groups. In the same time, this information must be provided extensively in the paper.
Materials and Methods
Please be more specific about the purpose of the research. What do you want to find, why do you conduct this research, with what this research contributes to the improvement of personal wellbeing of the future generation, etc. Would be useful if you will make a link between your purpose and the scientific literature. In the Discussion and conclusions you state that you analyzed three parameters, but you did not mention that in “Study design”.
In the same time, for a better understanding of the data collection you should make a presentation of the main functions of the application and how it works.
Discussion and conclusions
Managerial implications are not discussed at all in the paper. How your results can be used to improve the health policy in in the province of León or in Spain? Your findings will help the authorities to create public health programs. How the multi-platform web-based created by you can be used in the future?
Reviewer 2 Report
The study of an objective is to develop the effectiveness of BMI by the e-Health system for students. However, It is overstated in this content to difficultly read for audiences. I suggested the manuscript should be concise and write accurately. Generally, it can be published before the manuscript is improved as followed:
- The abstract should describe concisely and add quantitative outcomes, however, the conclusion should be short.
- The introduction section should also be concise and state the motivation and significance of the study. Why do you develop the e-Health system? Do you provide the niche of the system compared to previous studies? Do you want to mention the difference between other intervention methods and e-Health?
- The Materials and Methods section: Why do you mention the sources and comparability of the two groups (Tx and Control)? Does the protocol assess by IRB? I consider your study is a quasi-experimental design. How do you prevent the contamination of the control during the process? Do you provide the method to use the website in the intervention group? Do you measure the frequency of use e-Health and intensity of behavior change in Tx group? I suggest deleting the Figures 1 and 2. The statistical method should use the ACOVA and multivariate analysis to detect the difference between the two groups. The content should be concise.
- The Result section: All Tables and figures should be merged possibly and described precisely. The demographic and basic information should be compared and tested by statistical methods between the two groups. Do not separately describe the data in the two groups. All Data should be analyzed using multivariate analysis to adjusted other covariates.
- Conclusion section: Please discuss the niche and advantages of the e-Health system to promote the behavior change for the youth. Do you discuss problems of the system including contents, accessibility, availability, application for the vulnerable group, potential, limitation, etc? Regarding behavior change by e-health system, do you provide the theoretically support the finding? Do you point out the potential development of e-Health system? Do you feel the e-Health better than other health educational tools? What is the main reason and theory to support it?
-
Totally, please amend your manuscript concisely and try to focus
on the innovative ideas based on theory to academic field.
Round 2
Reviewer 2 Report
1. Totally, the manuscript has improved to illustration the unclear part. However, the manuscript must cutdown all reductant part and make it concisely as soon as possible. Such as, in the Introduction had classified into 10 parts, not focusing on the significance and motivation.
2. The statistical method was used to compare the differences by multivariate analysis, but you showed in Fig. was not compared in the two groups after adjusting for covariates. The Table 1 should compared the differences on data by independent T-test, not just show the data.
3. Please reorganize the whole contents to make it easily read and understand your findings, specially make the readable Table and Fig as soon as possible. You should concisely cutdown the contents to substitute to add more new data to illustrate.
4. The result should be rewrites clearly to explain the hypothesis of the study. In addition, the suitable theory was used to support the findings.
5. Generally, the manuscript was not easily understood what the main contribution was provided or highlighted. Please find the statistician to reanalysis the result and the display in the manuscript.
